# Measuring the impact of social-distancing, testing, and undetected asymptomatic cases on the diffusion of COVID-19

**Seungyoo Jeon** *

Department of Entrepreneurship and Small Business, Soongsil University, Seoul, Republic of Korea

* 1002228004@soongsil.ac.kr

## Abstract

The key to overcoming COVID-19 lies, arguably, in the diffusion process of confirmed cases. In view of this, this study has two main aims: first, to investigate the unique characteristics of COVID-19—for the existence of asymptomatic cases—and second, to determine the best strategy to suppress the diffusion of COVID-19. To this end, this study proposes a new compartmental model—the SICUR model—which can address undetected asymptomatic cases and considers the three main drivers of the diffusion of COVID-19: the degree of social distancing, the speed of testing, and the detection rate of infected cases. Taking each country's situation into account, it is suggested that susceptible cases can be classified into two categories based on their sources of occurrence: internal and external factors. The results show that the ratio of undetected asymptomatic cases to infected cases will, *ceteris paribus*, be 6.9% for South Korea and 22.4% for the United States. This study also quantitatively shows that to impede the diffusion of COVID-19: firstly, strong social distancing is necessary when the detection rate is high, and secondly, fast testing is effective when the detection rate is low.

## 1. Introduction

SARS-CoV-2 (COVID-19) has endangered the world. A remarkable aspect of COVID-19, unlike previous situations with viruses (e.g., MERS and SARS), is that it has been medically proven that asymptomatic cases exist—where infected people do not present any symptoms [1]. Such people can inadvertently and unconsciously transmit the virus to uninfected persons, although some asymptomatic cases can be detected if their paths cross with confirmed cases. However, this does not always happen, and many asymptomatic cases may go undetected.

To overcome the current global crisis, several studies suggest there may be a solution through mathematical modeling [2–4]. To analyze the COVID-19 outbreak, there have been many attempts to build models based on a classic compartmental model—the susceptible-infective-recovered (SIR) model developed by Kermack and McKendrick [5] -. Based on the SIR type model, most studies demonstrated the effect of a social distancing.

**Competing interests:** The author has declared that no competing interests exist.

According to the proposed compartment model (SEIHR), Choi and Ki [6] investigated the effectiveness of government interventions. Gounane et al. [7] addressed the effect of social distancing caused by public policy by introducing a new nonlinear SIR model. By adopting the time-varying infection rate, Cho and Kim [8] addressed the intervention effects of the events.

There was also research, not just on social distancing but also on detecting infected cases. Through analyzing the COVID-19 outbreak in its infancy, Anastassopoulou et al. [9] proposed a mathematical model based on the SIR model. Chen et al. [10] depicted the outbreak of COVID-19 at the initial stage by introducing a time-dependent SIR model. Samui et al. [11] suggested the SAIU model to investigate the effect of reporting infected cases. Ndairou et al. [12] developed the compartment model of COVID-19 with respect to the transmissibility of super-spreaders.

The following studies also considered the effect of speed of testing. Overton et al. [13] analyzed the effect of non-medical mediations in the early stage of COVID-19. Khan et al. [14] proposed the model considering undetected infected cases, social quarantine, release from quarantine, and re-infection. To manipulate the COVID-19 outbreak in the US, Tsay et al. [15] suggested an optimization-based decision-making strategy.

There was a research that dealt with the many waves of COVID-19. To investigate the spread of COVID-19 within a community, Cooper et al. [16] provided a theoretical framework based on the time-varying size of susceptibility. By applying a time-varying transmission rate, Gustavo et al. [17] considered sociological changes, including the change in the degree of social distancing and multiple waves of COVID-19. By introducing an SEIR(D) model, Shin [18] reflected the time-varying infection process of COVID-19 and the effectiveness of government intervention. The model proposed by Perakis et al. [19] reflected the time-varying population behavior with multi-waves.

Gaeta [20] showed that the SIR model is not appropriate to reflect the unique features of COVID-19, and it can be overcome by the modified model reflecting the existence of asymptomatic infectives. Notably, Gaeta [20] demonstrated not only the multi-waves of COVID-19 but also the detection of infected cases. Lee et al. [21] investigated the importance of public health interventions through control measures (quarantine and isolation). Lee et al. [21] demonstrated that public health intervention was crucial for (1) tackling the multi-waves of COVID-19 and (2) implementing speed of testing. Ramos et al. [22] reflected the effects of the various control measures (social distancing, contact tracing, and health interventions) with the multi-waves of COVID-19 by developing the modified SIR model. To consider the global dynamics of infection, AlQadi and Bani-Yaghomb [23] developed the extended SIR model. The above-mentioned studies are summarized in Table 1.

To encompass all the aforementioned features, this study proposes a new compartmental model–the SICUR model—which can address undetected asymptomatic cases. The model comprises five stages: susceptible cases (S), infected cases (I), confirmed cases (C), undetected asymptomatic cases (U), and recovered cases (R), as shown in Fig 1.

Notably, the term (I) refers to infected cases, not the "infectious cases" of the SIR model. The infected cases are composed of infectious, confirmed, and recovered cases. Based on the proposed model, this study demonstrates the impact of the degree of social distancing and the speed of testing, with different detection rates of infected cases, on the diffusion of COVID-19.

In applying the model to analyze the diffusion of COVID-19, this study assumes: First, that new births and deaths from a given susceptible case are ignored. Second, that recovered cases cannot be re-infected. Third, that the next step of confirmation (e.g., recovery or death) is not taken into account because the confirmed cases, regardless of the next status, cannot infect others. Fourth, that the latent period of COVID-19 (the duration of exposed individuals becoming infected) is neglected because the periods for asymptomatic cases are unobservable.

**Table 1. Description of SIR type models for COVID-19.**

| Authors | Data set | | Feature | | | | |
|---|---|---|---|---|---|---|---|
| | Countries | Observation period | Social distancing | Detection rate | Speed of Testing | Multi-waves | External wave |
| Choi and Ki (2020) [6] | Daegu, North Gyeongsang Province in South Korea | January 20, 2020 ~ March 4, 2020 | Y | | | | |
| Gounane et al. (2021) [7] | Germany, Spain, Italy, France, Algeria, Morocco | January 20, 2020 ~ July 14, 2020 | Y | | | | |
| Cho and Kim (2021) [8] | South Korea | January 20, 2020 ~ October 20, 2020 | Y | | | | |
| Anastassopoulou et al. (2020) [9] | Hubei in China | January 11, 2020 ~ February 10, 2020 | Y | Y | | | |
| Chen et al. (2020) [10] | USA, UK, France, Iran, Spain, Italy, Germany, South Korea | January 15, 2020 ~ March 2, 2020 | Y | Y | | | |
| Samui et al. (2020) [11] | India | January 30, 2020 ~ April 30, 2020 | Y | Y | | | |
| Ndairou et al. (2020) [12] | Wuhan in China | January 4, 2020 ~ March 9, 2020 | Y | Y | | | |
| Overton et al. (2020) [13] | Wuhan in China | December 1, 2019 ~ February 9, 2020 | Y | Y | Y | | |
| Khan et al. (2020) [14] | 8 states in the USA | January 22, 2020 ~ June 29, 2020 | Y | Y | Y | | |
| Tsay et al. (2020) [15] | USA | January 22, 2020 ~ April 16, 2020 | Y | Y | Y | | |
| Cooper et al. (2020) [16] | China, South Korea, India, Australia, USA, Italy, Texas in the USA | January, 2020 ~ June, 2020 | Y | | | Y | |
| Gustavo et al. (2021) [17] | Italy, Spain, USA | March 20, 2020 ~ November 15, 2020 | Y | | | Y | |
| Shin (2021) [18] | South Korea | February 18, 2020 ~ February 8, 2021 | Y | | | Y | |
| Perakis et al. (2022) [19] | All states in the USA | April 12, 2020 ~ February 15, 2021 | Y | | | Y | |
| Gaeta (2020) [20] | (Northern) Italy | February 21, 2020 ~ May 15, 2020 | Y | Y | | Y | |
| Lee et al. (2021) [21] | South Korea | January 20, 2020 ~ April 2, 2020 | Y | | Y | Y | |
| Ramos et al. (2021) [22] | Italy | January 19, 2020 ~ July 21, 2020 | Y | Y | Y | Y | |
| AlQadi and Bani-Yaghomb (2022) [23] | 6 cities and states in the USA | March 10, 2020 ~ March 7, 2021 | Y | | | Y | Y |
| Proposed Model | South Korea, USA | January 20, 2020 ~ December 31, 2020 | Y | Y | Y | Y | Y |

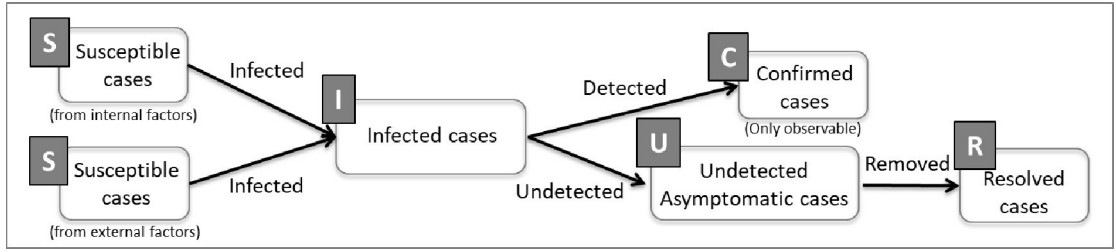

**Fig 1. Structure map of the SICUR model.**

Fifth, that the time of infection of immigrants, confirmed during COVID-19 testing, on entry, or during the self-quarantine period, is their entry time because the actual time of their infection is unobservable. Finally, that the estimation of the model is based on the number of confirmed cases because it is the only observable stage in the SICUR model.

## 2. Materials and methods

### 2.1 Model framework

The notations used in the proposed model are described in Table 2.

The SIR model is as follows.

$$\frac{dS}{dt} = -\frac{\beta IS}{N},$$

$$\frac{dI}{dt} = \frac{\beta IS}{N} - \gamma I, \tag{1}$$

$$\frac{dR}{dt} = \gamma I,$$

where $N = S + I + R$, and it is composed of three compartments; $S$ (the number of susceptible cases), $I$ (the number of infectious cases), and $R$ (the number of removed cases). The standard SIR model assumes that the entire population in a given country is susceptible at the initial time. "Never infected cases" exist—these are individuals who live through the pandemic without infection—and this happens many times. The power of infectivity may be underestimated through this. Therefore, accurate estimation must assume the realistic epidemic size of susceptible cases; and this study calls it the epidemic size $M$; $N = M$. Unlike the infectious cases $I$, $M(t)$ represents the infected cases, and the infected cases include not only infectious cases but also removed cases; $M(t) = I + R$. The removed cases can include the confirmed cases and the resolved cases; $R = N(t) + R(t)$. Then, the previous equation can be modified to,

$$\frac{dM(t)}{dt} = \frac{dI}{dt} + \frac{dR}{dt} = \frac{\beta IS}{N} = \frac{\beta}{M}(M(t) - N(t) - R(t))(M - M(t)), \tag{2}$$

since $S = N - I - R = M - M(t)$.

There are two main causes of rapid growth in the number of confirmed cases—a mass infection from super-spreaders, and the occurrence of major events (e.g., climate change, the beginning of a new semester, the announcement of the development of vaccines against COVID-19, or an intentional change in social-distancing regulations.)

First, a mass infection from super-spreaders is assumed to be the beginning of a subsequent wave of COVID-19 if the time to start the mass infection is after the time of the peak of the latest wave, and the time to start is the earliest date for which the number of confirmed cases in the next four days becomes at least double that of the previous four days for at least four consecutive days. Otherwise, mass infection from super-spreaders is assumed to be covered by the existing waves of COVID-19.

Second, the degree of social distancing and the epidemic size can be shifted when a major event occurs spontaneously. Since artificial operations are limited to changing the pool of susceptible cases, this study assumes that the epidemic size is fixed if the social-distancing regulations are lifted or relaxed; the degree of social distancing can be shifted without changing the epidemic size when a major event occurs intentionally.

To apply the causes into the model, this study adjusts Eq (2) as follows.

**Table 2. Notations.**

| Notation | Description | Formula |
|---|---|---|
| $M_{1,k}(t)$ | The cumulative number of infected cases from the $k$-th wave of COVID-19 at time $t$ ($k \geq 1$) | |
| $m_{1,k}(t)$ | The point-wise number of infected cases from the $k$-th wave of COVID-19 at time $t$ ($k \geq 1$) | $m_{1,k}(t) = \frac{dM_{1,k}(t)}{dt}$ |
| $M_2(t)$ | The cumulative number of infected cases from the rapid global diffusion of COVID-19 at time $t$ | |
| $m_2(t)$ | The point-wise number of infected cases from the rapid global diffusion of COVID-19 at time $t$ | $m_2(t) = \frac{dM_2(t)}{dt}$ |
| $M_{1,k}$ | The default epidemic size of susceptible cases from the $k$-th wave of COVID-19 ($k \geq 1$); the upper bound of $M_{1,k}(t)$ for all $t$ | |
| $M_2$ | The default epidemic size of susceptible cases from the rapid global diffusion of COVID-19; the upper bound of $M_2(t)$ for all $t$ | |
| $l_{M,t_s}$ | The multiplier for shifting the epidemic size at time $t_s$ ($s \geq 1$) | |
| $M_{1,k,t}$ | The epidemic size of susceptible cases from the $k$-th wave of COVID-19 at time $t$ ($k \geq 1$) | $M_{1,k,t} = \left( \prod_s l_{M,t_s} \right) M_{1,k}$ ($t_s \leq t$ for all s) |
| $M_{2,t}$ | The epidemic size of susceptible cases from the rapid global diffusion of COVID-19 at time $t$ | $M_{2,t} = \left( \prod_s l_{M,t_s} \right) M_2$ ($t_s \leq t$ for all s) |
| $M$ | The epidemic size of total susceptible cases of COVID-19, equal to or smaller than the national population | $M = \lim_{t \to \infty} \sum_k M_{1,k,t} + \lim_{t \to \infty} M_{2,t}$ |
| $N_{1,k}(t)$ | The cumulative number of confirmed cases from the $k$-th wave of COVID-19 at time $t$ ($k \geq 1$) | |
| $n_{1,k}(t)$ | The point-wise number of confirmed cases from the $k$-th wave of COVID-19 at time $t$ ($k \geq 1$) | $n_{1,k}(t) = \frac{dN_{1,k}(t)}{dt}$ |
| $N_2(t)$ | The cumulative number of confirmed cases from the rapid global diffusion of COVID-19 at time $t$ | |
| $n_2(t)$ | The point-wise number of confirmed cases from the rapid global diffusion of COVID-19 at time $t$ | $n_2(t) = \frac{dN_2(t)}{dt}$ |
| $N(t)$ | The cumulative number of confirmed cases at time $t$ | $N(t) = \sum_k N_{1,k}(t) + N_2(t)$ |
| $n(t)$ | The point-wise number of confirmed cases at time $t$ | $n(t) = \frac{dN(t)}{dt}$ |
| $R_{1,k}(t)$ | The cumulative removed number of undetected asymptomatic cases from the $k$-th wave of COVID-19 at time $t$ ($k \geq 1$) | |
| $r_{1,k}(t)$ | The point-wise removed number of undetected asymptomatic cases from the $k$-th wave of COVID-19 at time $t$ ($k \geq 1$) | $r_{1,k}(t) = \frac{dR_{1,k}(t)}{dt}$ |
| $R_2(t)$ | The cumulative removed number of undetected asymptomatic cases from the rapid global diffusion of COVID-19 at time $t$ | |
| $r_2(t)$ | The point-wise removed number of undetected asymptomatic cases from the rapid global diffusion of COVID-19 at time $t$ | $r_2(t) = \frac{dR_2(t)}{dt}$ |
| $q_{1,k}$ | The default rate of infection from the $k$-th wave of COVID-19 ($k \geq 1$) | |
| $q_2$ | The default rate of infection from the rapid global diffusion of COVID-19 | |
| $l_{q,t_s}$ | The multiplier for shifting the rate of infection at time $t_s$ ($s \geq 1$) | |
| $q_{1,k,t}$ | The rate of infection from the $k$-th wave of COVID-19 ($k \geq 1$) at time $t$ | $q_{1,k,t} = \left( \prod_{s=1} l_{q,t_s} \right) q_{1,k}$ ($t_s \leq t$ for all s) |
| $q_{2,t}$ | The rate of infection from the rapid global diffusion of COVID-19 at time $t$ | $q_{2,t} = \left( \prod_{s=1} l_{q,t_s} \right) q_2$ ($t_s \leq t$ for all s) |
| $A_{1,k}$ | The detection rate of infected cases from the $k$-th wave of COVID-19 ($k \geq 1$) | |
| $A_2$ | The detection rate of infected cases from the rapid global diffusion of COVID-19 | |
| $A$ | The detection rate of infected cases | |
| $c$ | The number of infected cases when the first confirmed case is detected | $c = M_{1,1}(1)$ |
| $I_{med}$ | The median of the virus shedding duration $I$ | |
| $\alpha$ | The shape parameter of the candidate for the distribution of the duration $I$ | |
| $\beta$ | The scale parameter of the candidate for the distribution of the duration $I$ | |
| $I_0$ | The duration of virus shedding between the time to be infected and the time to be resolved | |
| $\lambda_{1,k}$ | The removal rate for undetected asymptomatic cases from the $k$-th wave of COVID-19 | |
| $\lambda_2$ | The removal rate for undetected asymptomatic cases from the rapid global diffusion of COVID-19 | |

The rate of infection is compatible with the direct/indirect contact with infectious cases. The rate of infection can be accepted as the degree of social distancing; the lower rate of infection, the more effective social distancing. From Muller et al. [24], the Bass model was

motivated by compartmental models in epidemiology, such as the SIR model; the word-of-mouth effect and the remaining market potential in the Bass model correspond with the rate of infection and susceptible cases in the SIR model, respectively. From the generalized Bass model [25], the word-of-mouth effect and the market potential can be time-varying because of external influences. Hence, this study assumes that the rate of infection "$\beta$" and the epidemic size of susceptible cases "$M$" can be shifted; $\beta = q_t$, and $M = M_t$. Then, Eq (2) can be expressed as,

$$\frac{dM(t)}{dt} = \frac{q_t}{M_t}(M(t) - N(t) - R(t))(M_t - M(t)). \tag{3}$$

When there is only a single wave of COVID-19 because of the internal factors, Eq (3) becomes

$$\frac{dM_1(t)}{dt} = \frac{q_{1,t}}{M_{1,t}}(M_1(t) - N_1(t) - R_1(t))\big(M_{1,t} - M_1(t)\big), \tag{4}$$

where $M_1(t)$ and $M_{1,t}$ are the infected cases and the epidemic size in the first wave, respectively. When a major event occurs spontaneously at time $t_s$ ($s \geq 1$), the rate of infection is shifted by $l_{q,t_s}$, and the epidemic size is shifted by $l_{M,t_s}$. The spread of the virus will be more intensive when $l_{q,t_s} > 1$, and will be more widespread when $l_{M,t_s} > 1$. When a major event occurs intentionally at time $t_s$ ($s \geq 1$), the rate of infection is shifted by $l_{q,t_s}$ without changing the epidemic size.

From Peres et al. [26], cross-country influences can be multidimensional, and cross-country effects can be the consequence of "weak ties." Weak ties are due to the communication between adopters in one country and nonadopters in other countries. It can be further substantiated by Everdingen et al. [27]; the communication effect by previous adopters might result not only from someone within a population but also from across populations. It is applicable to the proposed model as follows. When a new additional wave is introduced because of the internal (or external) factors, susceptible cases in the new wave can be infected with the coronavirus not only by infectious individuals in the same wave but also by infectious individuals in the existing waves. This applies equally to susceptible cases in the existing waves. Hence, the number of virus spreaders can be expressed as the sum of all existing spreaders when the extra susceptible are added. For the $k$-th wave of COVID-19, the above equation is,

$$\frac{dM_k(t)}{dt} = \frac{q_{k,t}}{M_{k,t}}(M(t) - N(t) - R(t))\big(M_{k,t} - M_k(t)\big) * I(t \geq \tau_k), \tag{5}$$

where $q_{k,t}$ is the rate of infection from the $k$-th wave, and $\tau_k$ is the initial date on which the $k$-th wave of COVID-19 started. $M(t) = \sum_k M_k(t)$, where $M_k(t)$ is the infected cases from the $k$-th wave. $N(t) = \sum_k N_k(t)$, where $N_k(t)$ is the infected cases from the $k$-th wave. $R(t) = \sum_k R_k(t)$, where $R_k(t)$ is the resolved cases from the $k$-th wave. Since the confirmed cases are quarantined, and the resolved cases cannot infect others, the actual number of virus spreaders is $M(t) - N(t) - R(t)$. The actual number of susceptible residual cases is $M_{k,t} - M_k(t)$ and the probability that someone who comes in contact with a virus spreader is a residual susceptible case is $\frac{M_{k,t} - M_k(t)}{M_{k,t}}$. To reduce $\frac{dM_k(t)}{dt}$, the number of new infected cases—there are two strategies: First, to detect as many infectious cases as possible, and second to strengthen social distancing by decreasing $q_{k,t}$; in fact $q_{k,t}$ will be equal to zero under lockdown. If those control strategies work well for the $k$-th wave, the infected cases will decrease to a number lower than the final number of susceptible cases.

To take the particular situation of each country into account, this study suggests having two kinds of susceptible cases, based on the source of COVID-19: susceptible cases due to internal factors and external factors. To take account of rapid growth in the number of confirmed cases [28], this study suggests additional waves of COVID-19 and the expansion of existing waves. The total epidemic size is composed of both kinds of susceptible cases [29, 30].

**2.1.1 Susceptible cases from internal factors becoming infected cases.** Most confirmed cases are detected in a community, for example, the infection in and the spread from religious and social welfare facilities, and can be ascribed to internal factors. To reflect, the multiple waves of COVID-19 and the infections emanating from infectious cases, the point-wise number of infected cases from the $k$-th wave of COVID-19 can be defined as follows.

$$m_{1,k}(t) = \frac{dM_{1,k}(t)}{dt} = \frac{q_{1,k,t}}{M_{1,k,t}}(M(t) - N(t) - R(t))(M_{1,k,t} - M_{1,k}(t)) * \boldsymbol{I}(t \geq \tau_{1,k}), \qquad (6)$$

where $M(t) = \sum_k M_{1,k}(t) + M_2(t)$, $N(t) = \sum_k N_{1,k}(t) + N_2(t)$, $R(t) = \sum_k R_{1,k}(t) + R_2(t)$, and $\tau_{1,k}$ is the initial date on which the $k$-th wave of COVID-19 started.

**2.1.2 Susceptible cases from external factors becoming infected cases.** The spread of COVID-19 in any particular country is initiated by the infected cases abroad, some of which bring the virus into the country: In a declaration, the World Health Organization (WHO) has declared COVID-19 a pandemic [31]. This declaration and a sharp increase in the number of confirmed cases caused many people residing overseas to attempt to return to their homelands, and this caused an unnecessary infection. It is possible that countries whose governments did not restrict entry from abroad experienced a large influx of returning citizens.

Restrictions—such as if immigrants were confirmed to be infected after testing on entry, they were transferred to the hospital; and if not, they were advised to self-quarantine after their return from abroad—were not enforced by the governments of some countries. Immigrants in the latent period of infection could infect others sooner or later, thereby acting as spreaders of COVID-19.

These infections are differentiated from the infections of the existing susceptible cases, and therefore these cases can be defined as susceptible to external factors.

As with the conclusions from Eq (6), the point-wise number of infected cases from the rapid global diffusion of COVID-19 can be defined as follows,

$$m_2(t) = \frac{dM_2(t)}{dt} = \frac{q_{2,t}}{M_{2,t}}(M(t) - N(t) - R(t))(M_{2,t} - M_2(t)) * I(t \geq \tau_2), \qquad (7)$$

where $\tau_2$ is the initial date on which the rapid global diffusion of COVID-19 started.

**2.1.3 Infected cases becoming confirmed cases.** Regardless of whether there is an onset of symptoms, all immigrants who comply with the recommendations of the government can be detected by testing on entry or during the self-quarantine period. Excepting them, the infected cases not yet confirmed can be classified into two groups: detectable cases and undetectable cases. This study also assumes that no infected cases display any symptoms but are not subjected to any test, i.e., all symptomatic infected cases are tested. Hence, the pre-symptomatic infected cases can be detected because symptoms are eventually displayed. Since the asymptomatic cases, by definition, do not display any symptoms, it is assumed that they are not tested and thus cannot be confirmed (except for cases whose flows of movement overlap with those of confirmed cases). Hence, the asymptomatic cases can be detected retrospectively since tests are performed when symptoms occur or when it is disclosed that the flows of movement overlap with those of confirmed cases. In other words, there are also infected cases that are not detected by the administration of COVID-19 tests because they lack any symptoms of

COVID-19, and/or it is not revealed that their flows of movement overlap with those of confirmed cases. The point-wise number of confirmed cases from the $k$-th wave of COVID-19 and the point-wise number of confirmed cases from the rapid global diffusion of COVID-19 can be defined as follows,

$$n_{1,k}(t) = \int_0^{14} A_{1,k} * m_{1,k}(t-\tau) * P(I=\tau)d\tau, \text{ and} \tag{8}$$

$$n_2(t) = \int_0^{14} A_2 * m_2(t-\tau) * P(I=\tau)d\tau. \tag{9}$$

This study assumes that the detection rates of infected cases are time-invariant. The detection rate relies on the volume of testing: the more testing there is, the lower the number of undetected asymptomatic cases. For more testing to be conducted, it is necessary to find more test subjects, which in turn means that contact tracing must work better. Hence, it is assumed that the detection rate depends on the level of effectiveness of contact tracing.

As shown in Fig 2, the distribution of the number of links attached to each node determines the heterogeneity of a network [32, 33]. The most heterogeneous of the different topologies is the scale-free network [34]. If the potential transmission route in a specific wave of COVID-19 is closer to the scale-free network, susceptible cases from the specific wave are composed of most cases with a few links and a few major hubs able to act as super-spreaders which are specific infectious cases with a level of transmissibility that makes them capable of infecting other susceptible cases. The more links the major hub has, the more connections can be quickly traced, and the more effective contact tracing is. Other things (e.g., the guidelines of the center for disease control, the technological level, the capacity of tracing, and privacy issues) being equal in a single country, the effectiveness of contact tracing thus depends on the heterogeneity of a specific network. Therefore, it is assumed that the detection rate depends on the type of network within the specific wave of COVID-19. The detection rate can be regarded as a measure of how well contact tracing can work in the specific wave of COVID-19.

$P(I=\tau)$ is the probability that the duration $I$ of virus shedding (between the time to be infected and the time to be confirmed) is equal to $\tau$; the duration $I$ represents the speed of testing. Then, Eqs (8) and (9) mean that the infected cases from $M_{1,k}$ (or $M_2$) at time $t-\tau$ are confirmed after time $\tau$, and the time to be confirmed is $t$. Hence, this can be expressed as the convolution of $A_{1,k} * m_{1,k}(t-\tau)$ (or $A_2 * m_2(t-\tau)$) and $P(I=\tau)$. In addition, this study assumes that $P(I=\tau)$ for susceptible cases from external factors coincides with that for susceptible cases from internal factors, since the duration $I$ is homogenously distributed regardless of the type of susceptible case.

Based on estimates of the upper bounds of the COVID-19 incubation period, a period of 14 days is recommended for quarantining people who have had contact with a confirmed case [35]. In addition, the latest time of onset of symptoms is the latest time confirmed if the symptoms are unobservable [36]. Confirmed cases are those in which cases have become infected within the previous 14 days. Since the immigrants confirmed positive for COVID-19, whether on entry or during the self-quarantine period are completely isolated from immigration to confirmation, they cannot spread the virus; the duration of virus shedding in these events can be regarded as zero days. Hence, this study assumes that the virus shedding duration $I$ ranges from zero to fourteen days.

**2.1.4 Infected cases as undetected asymptomatic cases becoming resolved cases.** As with the conclusions from Eqs (8) and (9), the removed number of undetected asymptomatic cases from the $k$-th wave of COVID-19 and from the rapid global diffusion of COVID-19 can

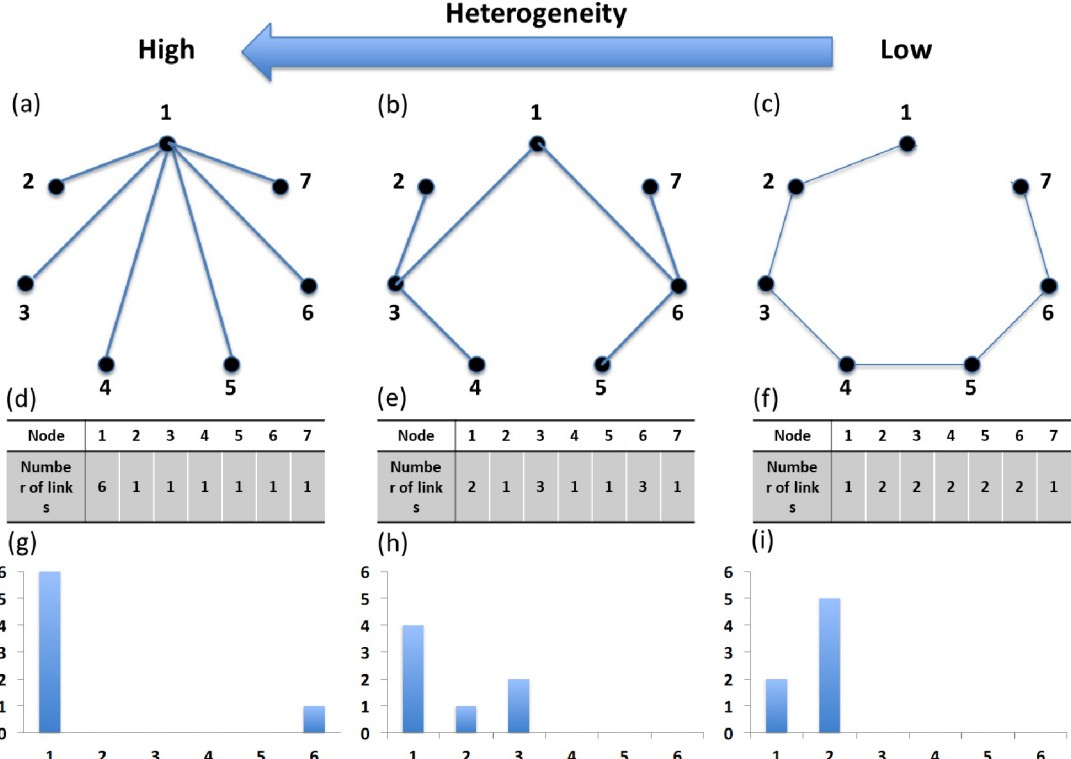

**Fig 2. Three kinds of connected graphs (the number of nodes = 7 / the sum of degrees = 12) are based on the heterogeneity of networks.** The example network with—(a) high heterogeneity. (b) intermediate heterogeneity. (c) low heterogeneity. The number of links for each node in—(d) the network A. (e) the network B. (f) the network C. The plot (x-axis = the number of links $r$ / the y-axis = the number of nodes with $r$ links) for—(g) the network A. (h) the network B. (i) the network C. There are three kinds of graphs based on the heterogeneity of the networks. If the detection of infected cases connected with detected cases is possible, all susceptible nodes (individuals) in the example network A (on the left side) can be detected for three periods at most. For example, node 2 is detected in period 1. Then, node 1, connected with node 2, can be detected in period 2. Since node 1 is connected with all the other nodes, all the left nodes can finally be detected in period 3. If the first detected node is node 1, all the susceptible nodes can be detected within two periods. However, all susceptible nodes in the example network C (on the right side) can be detected for at least four periods. For example, node 4 is detected in period 1, fortunately. Then, the nodes connected with node 4 (nodes 3 and 5) can be detected in period 2. Similarly, nodes 2 and 6 can be detected in period 3. Finally, nodes 1 and 7 can be detected in period 4. If the first detected node is not node 4, the number of periods required to detect all susceptible cases is more than five.

be defined as

$$r_{1,k}(t) = \int_0^t \left(1 - A_{1,k}\right) * m_{1,k}(t - \tau) * P_0(I_0 = \tau)d\tau, \text{ and} \tag{10}$$

$$r_2(t) = \int_0^t \left(1 - A_2\right) * m_2(t - \tau) * P_0(I_0 = \tau)d\tau. \tag{11}$$

Since the undetected asymptomatic cases are unobservable, this study directly addresses the removed infected cases without confirmation. $P_0(I_0 = \tau)$ is the probability that the duration $I_0$ of virus shedding is equal to $\tau$. Then, Eqs (10) and (11) mean that the infected cases from $M_{1,k}$ (or $M_2$) at time $t - \tau$ are removed after time $\tau$ without detection, and the time to be resolved is $t$. Hence, this can be expressed as the convolution of $(1 - A_{1,k}) * m_{1,k}(t - \tau)$ (or $(1 - A_2) * m_2(t - \tau)$) and $P_0(I_0 = \tau)$. In accordance with the assumption above, this study assumes that

$P_0(I_0 = \tau)$ for susceptible cases from external factors coincides with the probability of duration for susceptible cases from internal factors.

## 2.2 Data

The data consist of the daily number of confirmed cases in South Korea from January 20 to December 31, 2020, and the daily number of confirmed cases in the United States from January 22 to December 31, 2020. After the first confirmation of COVID-19 in South Korea (January 20, 2020), the number of daily cases was disclosed to the public by the Korean National Institute of Health (https://coronaboard.kr/en). After the first confirmation of COVID-19 in the United States (January 22, 2020), the number of daily cases was disclosed to the public by *Our World in Data* (https://ourworldindata.org/coronavirus-source-data).

## 2.3 Model fitting

### 2.3.1 Susceptible cases becoming infected cases.

$$m_{1,k}(t) = M_{1,k}(t+1) - M_{1,k}(t), \text{ and} \tag{12}$$

$$m_2(t) = M_2(t+1) - M_2(t). \tag{13}$$

For ease of calculation, Eqs (12) and (13) convert the continuous time to discrete time. Hence, the cumulative number of infected cases at time $t$, $M(t)$ is calculated as follows.

$$M(t) = \sum_k M_{1,k}(t) + M_2(t), \tag{14}$$

where $M_{1,k}(t) = \sum_{t'=1}^{t} m_{1,k}(t')$, and $M_2(t) = \sum_{t'=1}^{t} m_2(t')$.

### 2.3.2 Infected cases becoming confirmed cases.

$$n_{1,k}(t) = A_{1,k} * \sum_{s=1}^{14} m_{1,k}(t-s) * P(I = s), \text{ and} \tag{15}$$

$$n_2(t) = A_2 * \sum_{s=1}^{14} m_2(t-s) * P(I = s). \tag{16}$$

For ease of calculation, Eqs (15) and (16) convert the continuous time to discrete time. In the same way, the cumulative number of confirmed cases at time $t$, $N(t)$ is calculated as follows.

$$N(t) = \sum_k N_{1,k}(t) + N_2(t), \tag{17}$$

where $N_{1,k}(t) = \sum_{t'=1}^{t} n_{1,k}(t')$, and $N_2(t) = \sum_{t'=1}^{t} n_2(t')$. The probability $P(I = s)$ can be estimated as follows.

$$P(I = s) = [F(s) - F(s-1)]/F(14), \tag{18}$$

where $F(s)$ is the cumulative distribution function (cdf) of the duration $I$, and is assumed to follow the Gamma, Weibull, and Lognormal distributions–candidates for the distribution of the duration $I$. Since it is assumed that the duration $I$ ranges from 0 to 14, the probability that the duration $I$ is equal to $s$, $P(I = s)$ should be truncated; the candidates for the distribution of the incubation period are truncated to 14 days.

**2.3.3 Infected cases as undetected asymptomatic cases becoming resolved cases.**

$$r_{1,k}(t) = \left(1 - A_{1,k}\right) * \sum_{s=1}^{[t]} m_{1,k}(t-s) * P_0(I_0 = s), \text{ and} \tag{19}$$

$$r_2(t) = (1 - A_2) * \sum_{s=1}^{[t]} m_2(t-s) * P_0(I_0 = s). \tag{20}$$

For ease of calculation, Eqs (19) and (20) convert the continuous time to discrete time. In the same way, the cumulative number of removed cases at time $t$, $R(t)$ is calculated as

$$R(t) = \sum_k R_{1,k}(t) + R_2(t), \tag{21}$$

where $R_{1,k}(t) = \sum_{t'=1}^{t} r_{1,k}(t')$, and $R_2(t) = \sum_{t'=1}^{t} r_2(t')$. The probability $P_0(I_0 = s)$ can be estimated with

$$P_0(I_0 = s) = [F_0(s) - F_0(s - 1)], \tag{22}$$

where $F_0(s)$ is the cumulative distribution function (cdf) of the duration $I_0$, and assumes that the duration $I_0$ is followed by the Geometric distribution. The median duration of virus shedding is 28 days for asymptomatic infected cases [37]. To take this into account, this study assumes that the removal rate for undetected asymptomatic cases, $\lambda_{1,k}$ (or $\lambda_2$), is not estimated, but instead fixed at the value; $\lambda_{1,k}$ (or $\lambda_2$) is adjusted to make the median duration of $I_0$ equal to 28.

Since the number of infected cases is unobservable, the parameters are estimated based on the confirmed cases as follows.

$$SSE = \sum_t [N(t) - Y(t)]^2, \tag{23}$$

where $SSE$ is the sum of squared errors, and $Y(t)$ is the actual number of cumulative confirmed cases at time $t$. The parameters are estimated based on the confirmed cases by the nonlinear least squares (NLS) via the SAS 9.2 MODEL procedure.

# 3. Results

## 3.1 Estimation

**3.1.1 South Korea.** In South Korea, the first confirmed case was detected on January 20, which can be regarded as the initial date on which the first wave of COVID-19 started; this wave was augmented by a specific super-spreading event at the Shincheonji Church of Jesus in Daegu on February 18. Due to the declaration of WHO, there were additional susceptible cases from March 11, when a rapid global diffusion of COVID-19 started. Over the preceding four days–May 3 to May 6 –the total number of confirmed cases was 26, but 68 cases were confirmed in the next four days–May 7 to May 10; the ratio is 2.6. From the above assumption, the second wave, triggered by a specific super-spreading event at a club in Itaewon, Seoul, can be regarded as having started on May 7. After this, the third wave, sparked by a cluster at Sarang-Jeil church in Seoul, can be regarded as having started on August 12. (Over the preceding four days–August 8 to August 11 –the total number of confirmed cases was 141, but 379 cases were confirmed in the next four days–August 12 to August 15; the ratio is 2.7.) On October 12, there was a major intentional event: the South Korean government announced that the social-distancing regulations would go down to stage 1. On November 10 (in Korean time) [38], there was a spontaneous major event, when Pfizer declared its vaccines more than 90% effective against COVID-19. On November 24, another major intentional event occurred: the

South Korean government announced that the social-distancing regulations would go up to stage 2. As of December 31, 2020, in South Korea, there had been four waves with three shifts in the degree of social distancing and one shift in the epidemic size. Using the proposed model followed by the various distributions as the virus shedding duration $I$, this study estimates parameters. The estimated results are shown in Fig 3 and Table 3.

All parameters are fitted using the SICUR model except for the removal rates for undetected asymptomatic cases; those are significantly estimated except for the number of infected cases when the first confirmed case was detected, $c$. In particular, the p-value of the estimated $c$ is 0.0656 for the Lognormal, 0.0659 for the Gamma, and 0.0627 for the Weibull distribution; the Weibull distribution shows better performance than the Gamma distribution in terms of the stability of estimation. Hence, the Weibull distribution has been chosen for this study as the baseline distribution of the virus shedding duration $I$.

For the Weibull distribution, the weighted average ratio of the detection rate of infected cases is 92.6%. Contrary to popular belief, the ratio of undetected asymptomatic cases to confirmed cases is somewhat low. There are a few reasons for this phenomenon. If someone is judged to be a confirmed case, the Korea National Institute of Health starts to check his/her movements over the preceding two days, after which it checks with possibly encountered people whether the confirmed case has, in fact, encountered them and alerts them. Since people who have met with a confirmed case are advised to be tested regardless of whether they display any symptoms, many asymptomatic cases can be confirmed.

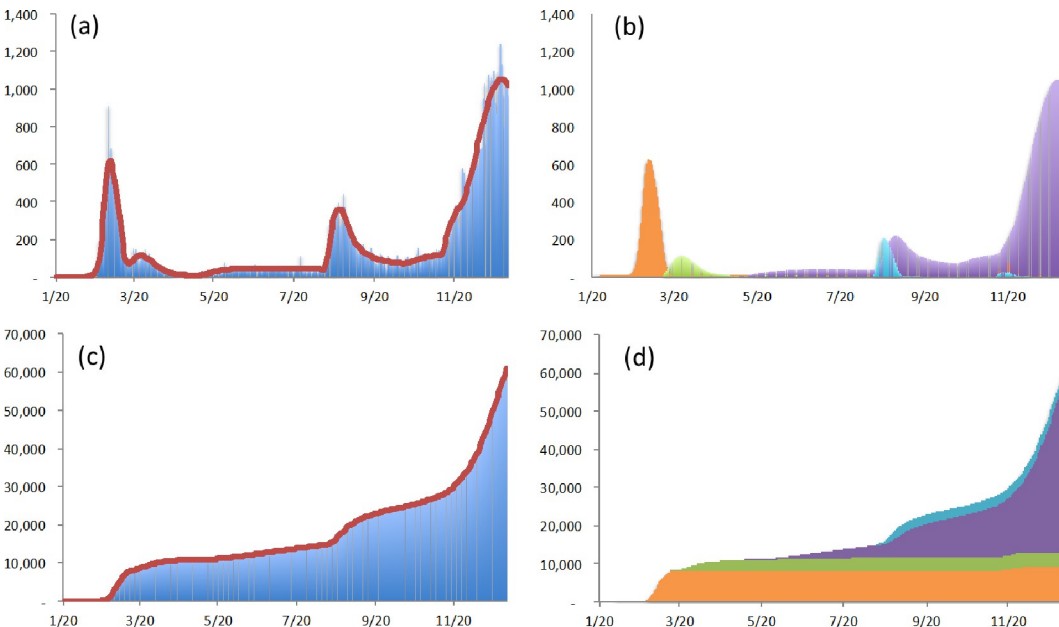

**Fig 3. Estimated numbers of confirmed cases in South Korea with the partition based on the sources of occurrence.** (a) The point-wise number of actual confirmed cases (blue vertical line) and the point-wise number of estimated confirmed cases (red solid line). (b) The point-wise number of estimated confirmed cases from the first wave (orange vertical line), the point-wise number of estimated confirmed cases from the rapid global diffusion (green vertical line), the point-wise number of estimated confirmed cases from the second wave (violet vertical line), and the point-wise number of estimated confirmed cases from the third wave (azure vertical line). (c) The cumulative number of actual confirmed cases (blue vertical line) and the cumulative number of estimated confirmed cases (red solid line). (d) The cumulative number of estimated confirmed cases from the first wave (orange vertical line), the cumulative number of estimated confirmed cases from the rapid global diffusion (green vertical line), the cumulative number of estimated confirmed cases from the second wave (violet vertical line), and the cumulative number of estimated confirmed cases from the third wave (azure vertical line).

**Table 3. Comparison of model fit and parameter estimates for confirmed cases in South Korea.**

| South Korea | Lognormal | | | Gamma | | | Weibull | | |
|---|---|---|---|---|---|---|---|---|---|
| | Estimate | Std err | p-value | Estimate | Std err | p-value | Estimate | Std err | p-value |
| $c$ | 0.0006 | 0.0003 | 0.0656 | 0.0005 | 0.0003 | 0.0659 | 0.0005 | 0.0003 | 0.0627 |
| $M_{1,1}$ | 8,518 | 234 | < .0001 | 8,489 | 228 | < .0001 | 8,482 | 226 | < .0001 |
| $M_2$ | 3,492 | 68 | < .0001 | 3,504 | 69 | < .0001 | 3,507 | 69 | < .0001 |
| $M_{1,2}$ | 102,528 | 3717 | < .0001 | 102,994 | 3763 | < .0001 | 103,102 | 3772 | < .0001 |
| $M_{1,3}$ | 2,491 | 46 | < .0001 | 2,489 | 46 | < .0001 | 2,489 | 46 | < .0001 |
| $l_{M,11/10}$ | 1.1181 | 0.0170 | < .0001 | 1.1170 | 0.0167 | < .0001 | 1.1167 | 0.0166 | < .0001 |
| $q_{1,1}$ | 0.6572 | 0.0260 | < .0001 | 0.6672 | 0.0258 | < .0001 | 0.6698 | 0.0251 | < .0001 |
| $q_2$ | 0.1887 | 0.0370 | < .0001 | 0.1934 | 0.0380 | < .0001 | 0.1946 | 0.0381 | < .0001 |
| $q_{1,2}$ | 0.1031 | 0.0035 | < .0001 | 0.1044 | 0.0038 | < .0001 | 0.1048 | 0.0038 | < .0001 |
| $q_{1,3}$ | 0.5260 | 0.0393 | < .0001 | 0.5172 | 0.0384 | < .0001 | 0.5149 | 0.0379 | < .0001 |
| $l_{q,10/12}$ | 1.3085 | 0.0149 | < .0001 | 1.3049 | 0.0152 | < .0001 | 1.3040 | 0.0152 | < .0001 |
| $l_{q,11/10}$ | 1.0321 | 0.0611 | < .0001 | 1.0352 | 0.0602 | < .0001 | 1.0360 | 0.0598 | < .0001 |
| $l_{q,11/24}$ | 1.5366 | 0.0984 | < .0001 | 1.5327 | 0.0969 | < .0001 | 1.5318 | 0.0964 | < .0001 |
| $A_{1,1}$[a] | 0.9602 | 0.0305 | < .0001 | 0.9624 | 0.0300 | < .0001 | 0.9630 | 0.0298 | < .0001 |
| $A_2$ | 0.9725 | 0.0404 | < .0001 | 0.9719 | 0.0390 | < .0001 | 0.9716 | 0.0385 | < .0001 |
| $A_{1,2}$ | 0.9180 | 0.0067 | < .0001 | 0.9191 | 0.0067 | < .0001 | 0.9193 | 0.0067 | < .0001 |
| $A_{1,3}$ | 1.0000 | 0.0000 | . | 1.0000 | 0.0000 | . | 1.0000 | 0.0000 | . |
| $\lambda$[b] | 0.9755 | 0.0000 | . | 0.9755 | 0.0000 | . | 0.9755 | 0.0000 | . |
| $\alpha$ | 2.6863 | 0.3444 | < .0001 | 1.3231 | 0.1231 | < .0001 | 1.2255 | 0.0769 | < .0001 |
| $\beta$ | 1.3671 | 0.1585 | < .0001 | 7.7513 | 1.5702 | < .0001 | 9.9804 | 1.2730 | < .0001 |
| $I_{med}$[c] | 5.665 | | | 5.628 | | | 5.616 | | |
| $M$ | 130,852 | | | 131,216 | | | 131,297 | | |
| $A$[d] | 0.9245 | | | 0.9255 | | | 0.9257 | | |
| $SSE$[e] | 1,527,224 | | | 1,528,709 | | | 1,529,460 | | |

[a] $A_{1,1}$ is restricted by the boundary condition ($0 \le A_{1,1} \le 1$).

[b] $\lambda = \lambda_{1,k}$ (or $\lambda_2$) such that $28 = \left\lceil \frac{-1}{\log_2(1-\lambda)} \right\rceil$.

[c] $I_{med}$ is the estimated median incubation period of COVID-19 (days).

[d] $A$ is the weighted mean of $A_{1,1}$, $A_{1,2}$, $A_{1,3}$, and $A_2$.

[e] SSE is the sum of squared error, and the adjusted R-squared values are 1.0000.

More specifically, the estimated detection rate for the first wave, $A_{1,1}$ is 0.96, but for the second wave, $A_{1,2}$ is 0.92, and for the third wave, $A_{1,3}$ is 1.00. This means that the underlying network types of the first and third waves are close to the scale-free network, but that of the second wave is not. A possible reason is that, although there was close interpersonal contact with a high degree of repeated exposure at the Shincheonji Church of Jesus in Daegu and at the Sarang-Jeil church in Seoul, these places did not cause the repeated exposure of the club in Itaewon, Seoul.

The estimated median time between infection and confirmation is about 5.6 days for the whole distribution of duration $I$. The estimated median incubation period (time from exposure to symptom onset) was 5.1 days, and the estimated median time from symptom onset to confirmation was 1.2 days [36]. The upper limit of the latent period ranged from zero to five days, with a median of one day [39]. Hence, the estimated duration I is broadly consistent with other estimates from previous studies, and the speed of testing in South Korea is comparatively acceptable.

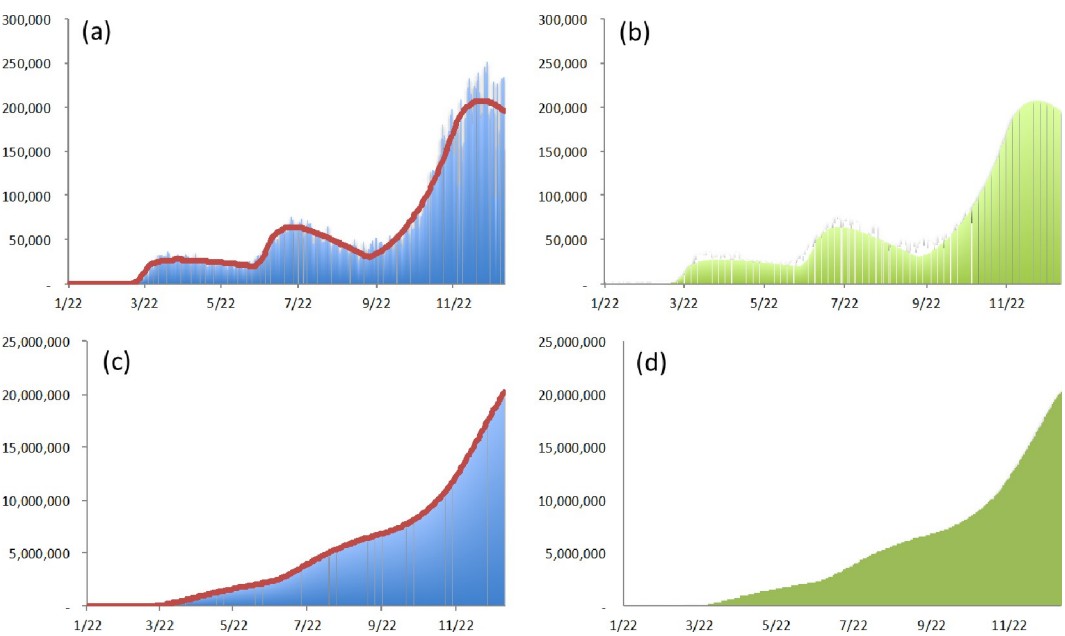

**Fig 4. Estimated numbers of confirmed cases in the United States with the partition based on the sources of occurrence.**
(a) The point-wise number of actual confirmed cases (blue vertical line) and the point-wise number of estimated confirmed cases (red solid line). (b) The point-wise number of estimated confirmed cases from the first wave (orange vertical line), and the point-wise number of estimated confirmed cases from the rapid global diffusion (green vertical line). (c) The cumulative number of actual confirmed cases (blue vertical line) and the cumulative number of estimated confirmed cases (red solid line). (d) The cumulative number of estimated confirmed cases from the first wave (orange vertical line), and the cumulative number of estimated confirmed cases from the rapid global diffusion (green vertical line).

In general, the degree of social distancing will be lower ($l_{q,t_s'} > 1$) if the social-distancing regulations are relaxed. This can be verified by the estimated multiplier for shifting the rate of infection ($l_{q,10/12} = 1.30$) on October 12. Similarly, the degree of social distancing will be higher ($l_{q,t_s'} < 1$) if the social-distancing regulations are lifted. However, the estimated multiplier for shifting the rate of infection on November 24, $l_{q,11/24}$ is 1.53, which means that the social-distancing regulations implemented on November 24 were not effective in preventing the spread of COVID-19.

When the development of vaccines was announced on November 10, people started to resume outdoor activities with the expectation that COVID-19 would soon end (despite no further details on the schedule of vaccination). Then the number of people one met increased and the duration of contact was extended, which means that the spread of the virus became more widespread ($l_{M,11/10} = 1.12$) and more intensive ($l_{q,11/10} = 1.04$).

**3.1.2 United States.** In the United States, the first confirmed case was detected on January 22, which can be regarded as the initial date on which the first wave of COVID-19 started. In accordance with South Korea, there were additional susceptible cases from March 11. In June, the number of confirmed cases more than doubled in 14 states because of businesses resuming against the recommendations of the National Institute of Health [40]. Hence, this study assumes that the first spontaneous major event occurred on June 15. Following that, the start of the fall semester was the second spontaneous major event, and it occurred on September 1 [41]. With the Pfizer Inc declaration on November 9 [38], Federal health officials announced an agreement to distribute vaccines (after approval) for free at pharmacies nationwide on November 12 [42]. This was the third spontaneous major event. As of December 31, 2020, in the United States, there had been two waves with three shifts in the degree of social distancing and three shifts in the epidemic size. The estimated results are shown in Fig 4 and Table 4.

**Table 4. Comparison of model fit and parameter estimates for confirmed cases in the United States.**

| USA | Lognormal | | | Gamma | | | Weibull | | |
|---|---|---|---|---|---|---|---|---|---|
| | Estimate | Std err | p-value | Estimate | Std err | p-value | Estimate | Std err | p-value |
| $c$ | 861,104 | 264,895 | 0.0013 | 702,643 | 187,043 | 0.0002 | 702,504 | 187,009 | 0.0002 |
| $M_{1,1}$ | 1,462,682 | 889,515 | 0.101 | 1,363,060 | 964,539 | 0.1585 | 1,362,692 | 964,221 | 0.1585 |
| $M_2$ | 2.23E+7 | 3.35E+6 | $< .0001$ | 1.94E+7 | 3.06E+6 | $< .0001$ | 1.94E+7 | 3.06E+6 | $< .0001$ |
| $l_{M,6/15}$ | 0.5993 | 0.1698 | 0.0005 | 0.7005 | 0.2190 | 0.0015 | 0.7005 | 0.2190 | 0.0015 |
| $l_{M,9/1}$ | 22.9763 | 5.0906 | $< .0001$ | 22.5275 | 5.8150 | 0.0001 | 22.5284 | 5.8144 | 0.0001 |
| $l_{M,11/12}$ | 0.3221 | 0.0431 | $< .0001$ | 0.3306 | 0.0542 | $< .0001$ | 0.3306 | 0.0542 | $< .0001$ |
| $q_{1,1}$ | 0.0268 | 0.0081 | 0.0011 | 0.0305 | 0.0102 | 0.0031 | 0.0305 | 0.0102 | 0.0031 |
| $q_2$ | 0.0548 | 0.0177 | 0.0021 | 0.0565 | 0.0221 | 0.011 | 0.0565 | 0.0221 | 0.011 |
| $l_{q,6/15}$ | 2.8447 | 0.4080 | $< .0001$ | 2.5806 | 0.2961 | $< .0001$ | 2.5804 | 0.2961 | $< .0001$ |
| $l_{q,9/1}$ | 0.3571 | 0.0655 | $< .0001$ | 0.3694 | 0.0719 | $< .0001$ | 0.3694 | 0.0719 | $< .0001$ |
| $l_{q,11/12}$ | 1.1479 | 0.0272 | $< .0001$ | 1.1675 | 0.0282 | $< .0001$ | 1.1676 | 0.0282 | $< .0001$ |
| $A_{1,1}$[a] | 0.0000 | 0.0000 | . | 0.0000 | 0.0000 | . | 0.0000 | 0.0000 | . |
| $A_2$[b] | 0.8298 | 0.0833 | $< .0001$ | 0.8305 | 0.0907 | $< .0001$ | 0.8306 | 0.0907 | $< .0001$ |
| $\lambda^{2)}$ | 0.9755 | 0.0000 | . | 0.9755 | 0.0000 | . | 0.9755 | 0.0000 | . |
| $\alpha$ | 151.0247 | 95221.3 | 0.9987 | 1.5809 | 0.3483 | $< .0001$ | 1.5806 | 0.3482 | $< .0001$ |
| $\beta$ | 8.8992 | 2834.9 | 0.9975 | 8.27E+69 | 8.20E-84 | $< .0001$ | 8.27E+69 | 4.77E-84 | $< .0001$ |
| $I_{med}$ | 9.6879 | | | 9.0304 | | | 9.0297 | | |
| $M$ | 106,297,276 | | | 108,280,631 | | | 108,272,647 | | |
| $A$[c] | 0.7787 | | | 0.7760 | | | 0.7760 | | |
| $SSE$[d] | 6.040E+11 | | | 6.577E+11 | | | 6.577E+11 | | |

[a]$A_{1,1}$ and $A_2$ are restricted by the boundary condition ($0 \leq A_{1,1}$, $A_2 \leq 1$).

[b]$\lambda = \lambda_{1,k}$ (or $\lambda_2$) such that $28 = \left\lceil \frac{-1}{log_2(1-\lambda)} \right\rceil$.

[c]$A$ is the weighted mean of $A_{1,1}$ and $A_2$.

[d]SSE is the sum of squared error, and the adjusted R-squared values are 0.9999.

As with the case of South Korea, all parameters are fitted using the SICUR model except for the removal rates for undetected asymptomatic cases; those are significantly estimated except for the default epidemic size of susceptible cases from the first wave of COVID-19, $M_{1,1}$. In accordance with South Korea's case, the Weibull distribution has been chosen in this study for the distribution of the duration $I$.

For the Weibull distribution, the weighted average ratio of the detection rate of infected cases is 77.6%. The undetected rate in the United States is higher than in South Korea. For all the candidates for the distribution of the duration $I$, the estimated median time between infection and confirmation is longer than 9.0 days; it can be expected that some infected cases are undetected even when symptoms occur because of the low capacity of the healthcare system in the United States.

When resuming business in mid-June, people began to crowd inside as the weather heated up outside. This was demonstrated by the reduced epidemic size ($l_{M,6/15} = 0.70$) and the more intensive spread of the virus ($l_{q,6/15} = 2.58$). When starting the fall semester with the cool weather, people started to resume outdoor activities which meant that despite the reduced duration of contact, the number of people one met increased; the spread of the virus became less intensive ($l_{q,9/1} = 0.37$), but the epidemic size was greatly expanded ($l_{M,9/1} = 22.53$). Compared with the case of South Korea, the announcement of developing vaccines brought about the opposite result that the spread of the virus became more intensive ($l_{q,11/12} = 1.17$), but the

epidemic size was reduced ($l_{M,11/12}$ = 0.33). Because the weather was getting cold, people were gathered indoors; people reacted more sensitively to the climate change than in anticipation of ending COVID-19.

The most remarkable aspect of the case of the United States is that the estimated $c$ is more than 700,000, implying that more than 700,000 infected cases could have remained undetected until the first confirmed case was announced in the United States. Moreover, the estimated $A_{1,1}$ –the detection rate of infected cases of the first wave–is almost zero; only a few infected cases were detected. With the entry of susceptible cases from external factors, the majority of infected cases remaining undetected became the trigger of the rapid growth in the number of confirmed cases.

## 3.2 Simulation

For South Korea, the estimated multiplier, $l_{q,10/12}$, for shifting the rate of infection on October 12 is 1.30, that is, the degree of social distancing worsened by 1.30 times as a result of the social-distancing policy announced on October 12. As shown in Fig 5, the cumulative number of confirmed cases as of December 31, 2020 will be 40,388 if there is no shift in the degree of social distancing. The artificial shifting of the degree of social distancing incurs an additional 20,346 confirmed cases as of December 31, 2020.

## 3.3 Prediction

As with the beginning of the second wave, it is highly probable that the next wave will be triggered by the long holidays. For the case of South Korea, this study assumes that there will be a fourth wave beginning May 1, 2021. This study verifies several scenarios by shifting three terms: the degree of social distancing (Low: $q_{1,4}$ = 0.4 / Mid: $q_{1,4}$ = 0.2 / High: $q_{1,4}$ = 0.1), the detection rate of infected cases (High: $A_{1,4}$ = 1.0 / Mid: $A_{1,4}$ = 0.75 / Low: $A_{1,4}$ = 0.5), and the speed of testing (High: $I_{med}$ = 3.7 / Mid: $I_{med}$ = 5.6 / Low: $I_{med}$ = 7.5). It is assumed that the final number of susceptible cases from internal factors is the same for all scenarios ($M_{1,4}$ = 10,000), and that the baseline model is the SICUR model followed by the Weibull distribution.

**3.3.1 Hypothesis 1: The stronger the social distancing, the lower the diffusion of COVID-19.**    To determine the impact of social distancing on the diffusion of COVID-19, this study considers three scenarios with different detection rates of infected cases, and different speeds of testing. The degree of social distancing for Scenario 2 is the default 0.2. The degree of

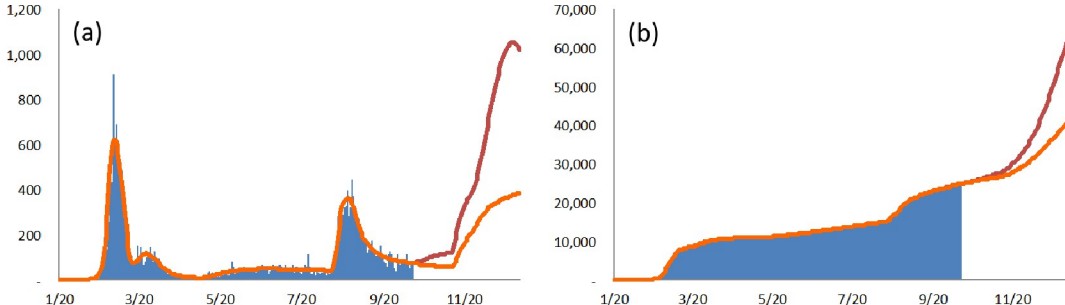

**Fig 5. Estimated numbers of confirmed cases in South Korea with/without the shift of $q$.** (a) The point-wise number of actual confirmed cases (blue vertical line), the point-wise number of estimated confirmed cases (red solid line), and the point-wise number of estimated confirmed cases without the shift of the degree of social distancing on October 12 (orange solid line). (b) The cumulative number of actual confirmed cases (blue vertical line), the cumulative number of estimated confirmed cases (red solid line), and the cumulative number of estimated confirmed cases without the shift of the degree of social distancing on October 12 (orange solid line).

**Table 5. Demonstration of the effects of social distancing.**

| Scenarios (1 / 2 / 3) | | $t_{peak,1,4}$[a] (days) | | | $n(t_{peak,1,4})$ | | |
|---|---|---|---|---|---|---|---|
| | | $q = 0.4$ | $q = 0.2$ | $q = 0.1$ | $q = 0.4$ | $q = 0.2$ | $q = 0.1$ |
| $A = 1.0$ | $I_{med} = 3.7$ | 11 | 17 | 20 | 876 | 447 | 178 |
| | $I_{med} = 5.6$ | 14 | 20 | 25 | 912 | 551 | 223 |
| | $I_{med} = 7.5$ | 15 | 21 | 28 | 981 | 638 | 270 |
| $A = 0.75$ | $I_{med} = 3.7$ | 12 | 20 | 32 | 739 | 443 | 206 |
| | $I_{med} = 5.6$ | 14 | 21 | 33 | 769 | 514 | 247 |
| | $I_{med} = 7.5$ | 16 | 22 | 33 | 827 | 574 | 283 |
| $A = 0.5$ | $I_{med} = 3.7$ | 12 | 21 | 38 | 586 | 417 | 248 |
| | $I_{med} = 5.6$ | 15 | 22 | 37 | 620 | 463 | 273 |
| | $I_{med} = 7.5$ | 16 | 23 | 37 | 675 | 503 | 295 |
| Default | | 0 | | | 45 | | |

[a]$t_{peak,1,4}$ is the time to peak of the fourth wave of COVID-19.

social distancing for Scenario 1 is 0.4, and the strength of transmissibility is twice the default, which means that people are paying less attention to social distancing. The degree of social distancing for Scenario 3 is 0.1, and the strength of transmissibility is half the default, which means that people are paying more attention to social distancing.

As shown in Table 5 and Fig 6, the half-strength of transmissibility (Scenario 3) reduces the peak number of confirmed cases regardless of the detection rate and the speed of testing. However, the double-strength of transmissibility (Scenario 1) increases the peak number of

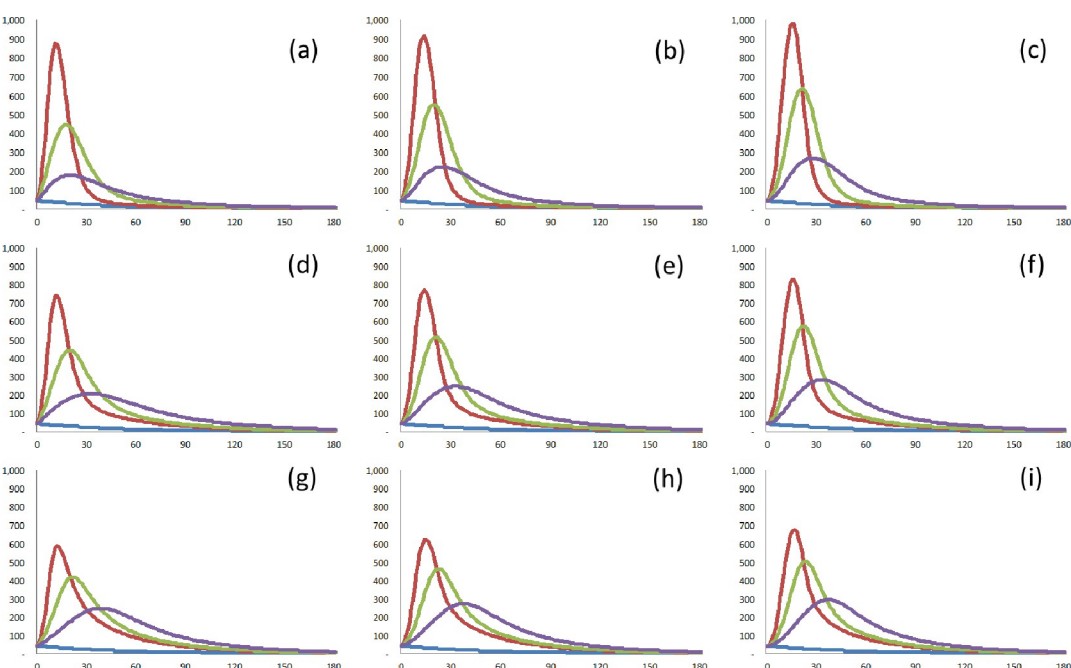

**Fig 6. Demonstration of the effects of social distancing.** The confirmed cases without a new wave (default; blue solid line), the confirmed cases with $q = 0.4$ (red solid line), the confirmed cases with $q = 0.2$ (green solid line), and the confirmed cases with $q = 0.1$ (violet solid line). (a) A = 1.0, $I_{med} = 3.7$. (b) A = 1.0, $I_{med} = 5.6$. (c) A = 1.0, $I_{med} = 7.5$. (d) A = 0.75, $I_{med} = 3.7$. (e) A = 0.75, $I_{med} = 5.6$. (f) A = 0.75, $I_{med} = 7.5$. (g) A = 0.5, $I_{med} = 3.7$. (h) A = 0.5, $I_{med} = 5.6$. (i) A = 0.5, $I_{med} = 7.5$.

**Table 6. Demonstration of the effects of detection rate.**

| Scenarios (1 / 2 / 3) | | $t_{peak,1,4}$ (days) | | | $n(t_{peak,1,4})$ | | |
|---|---|---|---|---|---|---|---|
| | | $A = 1.0$ | $A = 0.75$ | $A = 0.5$ | $A = 1.0$ | $A = 0.75$ | $A = 0.5$ |
| $q = 0.4$ | $I_{med} = 3.7$ | 11 | 12 | 12 | 876 | 739 | 586 |
| | $I_{med} = 5.6$ | 14 | 14 | 15 | 912 | 769 | 620 |
| | $I_{med} = 7.5$ | 15 | 16 | 16 | 981 | 827 | 675 |
| $q = 0.2$ | $I_{med} = 3.7$ | 17 | 20 | 21 | 447 | 443 | 417 |
| | $I_{med} = 5.6$ | 20 | 21 | 22 | 551 | 514 | 463 |
| | $I_{med} = 7.5$ | 21 | 22 | 23 | 638 | 574 | 503 |
| $q = 0.1$ | $I_{med} = 3.7$ | 20 | 32 | 38 | 178 | 206 | 248 |
| | $I_{med} = 5.6$ | 25 | 33 | 37 | 223 | 247 | 273 |
| | $I_{med} = 7.5$ | 28 | 33 | 37 | 270 | 283 | 295 |
| Default | | 0 | | | 45 | | |

confirmed cases for all cases, as with the above scenario. This means that effective social distancing can delay and reduce the diffusion of COVID-19.

**3.3.2 Hypothesis 2: The higher the detection rate, the lower the diffusion of COVID-19.** To determine the effect of the detection rate, this study considers three scenarios with different degrees of social distancing, and different speeds of testing. The detection rate of infected cases for Scenario 2 is 0.75: the detection rate is the default, implying that 25% of the infected cases would not be detected. The detection rate of infected cases for Scenario 1 is 1.0, which means that all the infected cases are fully detected. The detection rate of infected cases for Scenario 3 is 0.5, which means that half of the infected cases would not be detected.

It can be expected that the easier it is to trace the spread of COVID-19 in a specific wave, the more asymptomatic cases will be detected. If so, first, the sooner the time to peak for confirmed cases, and second, the higher the number of peak confirmed cases. As shown in Table 6 and Fig 7, it can be verified that the speed of diffusion increases as the detection performance improves; the time to peak $t_{peak,1,4}$ is delayed as the detection rate decreases. However, the magnitude of diffusion depends on the degree of social distancing; the number of peak confirmed cases $n(t_{peak,1,4})$ for $A_{1,4} = 1.0$ is at its highest when $q_{1,4} = 0.4$, but $n(t_{peak,1,4})$ for $A_{1,4} = 1.0$ is at its lowest when $q_{1,4} = 0.1$. Hence, the simulation results are partially in line with expectations. To impede the diffusion of a specific wave in which a few super-spreaders are to be expected, it is necessary to detect as many confirmed cases as possible. Although more detection may consume more time and resources, improving the detection of confirmed cases effectively curbs the spread of COVID-19.

**3.3.3 Hypothesis 3: The faster the testing, the lower the diffusion of COVID-19.** To measure the effect of the speed of testing, this study focuses on the number of peak confirmed cases $n(t_{peak,1,4})$ by shifting the median of the duration $I$ ($I_{med}$). $I_{med}$ is shifted by changing the scale parameter $\beta$. As with hypotheses 1 and 2, this study considers three scenarios with different degrees of social distancing, and different detection rates of infected cases.

The duration median, $I_{med}$, for Scenario 2 is the default 5.6 days: all symptomatic infected cases are tested shortly after symptoms occur, and some asymptomatic cases are detected only if the flow of movement overlaps with those of confirmed cases for the previous two days.

The median of the duration $I$ for Scenario 1 is 3.7 days: the speed of testing is on average 50% better than the default, which means that all symptomatic infected cases are tested as soon as symptoms occur (e.g., when doctors provide a proactive diagnosis rather than waiting for patients to visit); and more asymptomatic cases than the default are detected. (e.g., when cases

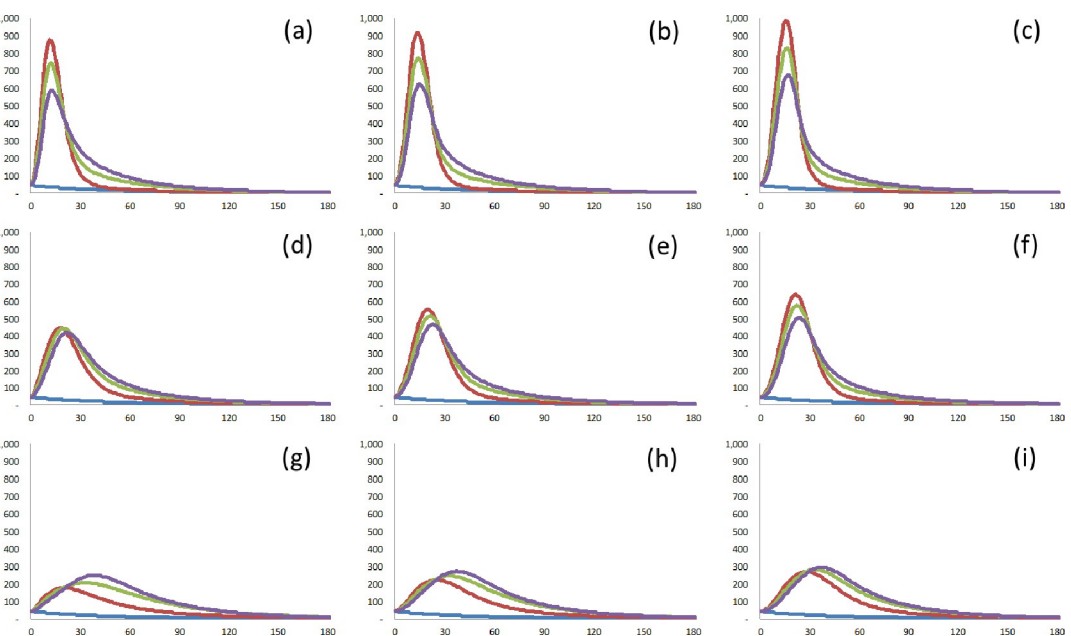

**Fig 7. Demonstration of the effects of detection rate.** The confirmed cases without a new wave (blue solid line), the confirmed cases with A = 1.0 (red solid line), the confirmed cases with A = 0.75 (green solid line), and the confirmed cases with A = 0.5 (violet solid line). (a) $q = 0.4$, $I_{med} = 3.7$. (b) $q = 0.4$, $I_{med} = 5.6$. (c) $q = 0.4$, $I_{med} = 7.5$. (d) $q = 0.2$, $I_{med} = 3.7$. (e) $q = 0.2$, $I_{med} = 5.6$. (f) $q = 0.2$, $I_{med} = 7.5$. (g) $q = 0.1$, $I_{med} = 3.7$. (h) $q = 0.1$, $I_{med} = 5.6$. (i) $q = 0.1$, $I_{med} = 7.5$.

are tested if the flow of movement overlaps with those of the confirmed cases for the previous three days or longer.)

The median of the duration $I$ for Scenario 3 is 7.5 days. The duration of virus shedding is on average 33% longer than the default, which means that the testing of the symptomatic infected cases may be delayed despite symptoms (e.g., only the serious cases are tested because of inadequate medical infrastructure); and most asymptomatic cases are not tested. (e.g., there is no pressure for the asymptomatic cases to be tested regardless of whether the flow of movement overlaps with those of confirmed cases).

From hypothesis 2, it can be concluded that the more asymptomatic cases are detected, the sooner the time to peak for confirmed cases. As shown in Table 7 and Fig 8, the shorter the

**Table 7. Demonstration of the effects of the speed of testing.**

| Scenarios (1 / 2 / 3) | | $t_{peak,1,4}$ (days) | | | $n(t_{peak,1,4})$ | | |
|---|---|---|---|---|---|---|---|
| | | $I_{med} = 3.7$ | $I_{med} = 5.6$ | $I_{med} = 7.5$ | $I_{med} = 3.7$ | $I_{med} = 5.6$ | $I_{med} = 7.5$ |
| $q = 0.4$ | $A = 1.0$ | 11 | 14 | 15 | 876 | 912 | 981 |
| | $A = 0.75$ | 12 | 14 | 16 | 739 | 769 | 827 |
| | $A = 0.5$ | 12 | 15 | 16 | 586 | 620 | 675 |
| $q = 0.2$ | $A = 1.0$ | 17 | 20 | 21 | 447 | 551 | 638 |
| | $A = 0.75$ | 20 | 21 | 22 | 443 | 514 | 574 |
| | $A = 0.5$ | 21 | 22 | 23 | 417 | 463 | 503 |
| $q = 0.1$ | $A = 1.0$ | 20 | 25 | 28 | 178 | 223 | 270 |
| | $A = 0.75$ | 32 | 33 | 33 | 206 | 247 | 283 |
| | $A = 0.5$ | 38 | 37 | 37 | 248 | 273 | 295 |
| Default | | 0 | | | 45 | | |

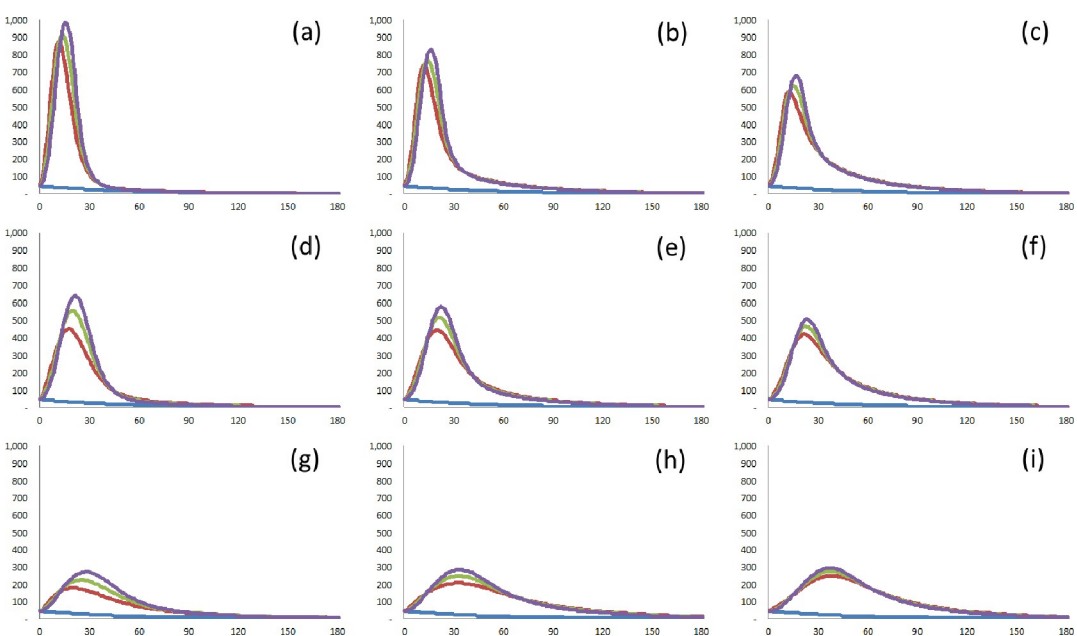

**Fig 8. Demonstration of the effects of the speed of testing.** The confirmed cases without a new wave (blue solid line), the confirmed cases with $I_{med}$ = 3.7 (red solid line), the confirmed cases with $I_{med}$ = 5.6 (green solid line), and the confirmed cases with $I_{med}$ = 7.5 (violet solid line). (a) $q$ = 0.4, A = 1.0. (b) $q$ = 0.4, A = 0.75. (c) $q$ = 0.4, A = 0.5. (d) $q$ = 0.2, A = 1.0. (e) $q$ = 0.2, A = 0.75. (f) $q$ = 0.2, A = 0.5. (g) $q$ = 0.1, A = 1.0. (h) $q$ = 0.1, A = 0.75. (i) $q$ = 0.1, A = 0.5.

duration $I$ is, first, the sooner the time to peak for confirmed cases, and second, the lower the number of peak confirmed cases. Unlike the situation with hypothesis 2, it can be verified that the peak point of the confirmed cases is on the upward slope. This means that improvements in the speed of testing can reduce the diffusion of COVID-19. In particular, the gap in the number of peak confirmed cases $n(t_{peak,1,4})$ decreases as the detection rate $A$ decreases (or as the degree of social distancing $q$ increases). This means that the more asymptomatic cases are detected (or the stronger the social distancing), the more effective the increased speed of testing. This corresponds to the speed of testing on the scale-free network being more important than on any other type of network [43]. Faster testing may also consume more time and resources, but improvements in the speed of testing are also effective in curbing the spread of COVID-19.

## 4. Discussion

Today, the existence of undetected asymptomatic cases of COVID-19 may no longer be surprising. However, it is still not clear how many undetected asymptomatic cases there are now. Unlike earlier viruses, the existence of undetected asymptomatic cases is the distinguishing feature of COVID-19, the previous compartmental models in epidemiology are limited in their ability to reflect and explain this phenomenon. To close this gap, this study proposes a new epidemiological model, the SICUR model.

This study has shown the effects of social distancing and a control system from South Korea and the United States. It is essential to measure the detection rate because the optimal strategy in preparing for the diffusion of COVID-19 may depend on whether contact tracing is effective for a specific wave. This can also be applied to determine vaccination priorities. The closer the contact to confirmed cases, the more likely the risk of infection. Initially, vaccinating people in their 70s and older, with a high mortality rate and close contact with confirmed

cases, will effectively suppress the spread of COVID-19 given limited vaccine availability. The process described in this study could be used to examine each country's system for dealing with COVID-19 based on the estimated degree of social distancing and the speed of testing. Accurate knowledge of the current level of prevention would be a key factor in the early elimination of COVID-19.

In this study, there are several limitations as follows.

1. Since the forecasting of the beginning time and the size of a new wave is beyond this study, it is unfeasible to estimate the time until the end of COVID-19, nor is it feasible to estimate the final number of confirmed cases at the end.

2. Until the end of 2020 –the final date of in-sample, vaccination had been rare in South Korea. If the effectiveness of vaccination is considered within the extended period of in-sample, this study can develop a more effective model.

3. The proposed model is inadequate in dealing with new mutations of COVID-19. (e.g., Omicron.) This study assumes that COVID-19 re-infection is unlikely to happen. In addition, this study is unable to verify the remarkable mutation of COVID-19, merely since it occurs within the period of in-sample.

4. Since this study focuses on only the confirmed cases, supplementary analysis is required to model the additional components (recovered cases or death cases) in the spread of infection.

The above mentioned limitations can be investigated with further analysis.

## Supporting information

**S1 Dataset.**
(XLSX)

## Acknowledgments

The author is grateful to Ki Sung Yang (Soongsil University, Seoul, Republic of Korea) for his helpful comments to enhance the quality of this study.

## Author Contributions

**Conceptualization:** Seungyoo Jeon.

**Data curation:** Seungyoo Jeon.

**Formal analysis:** Seungyoo Jeon.

**Funding acquisition:** Seungyoo Jeon.

**Investigation:** Seungyoo Jeon.

**Methodology:** Seungyoo Jeon.

**Project administration:** Seungyoo Jeon.

**Resources:** Seungyoo Jeon.

**Software:** Seungyoo Jeon.

**Supervision:** Seungyoo Jeon.

**Validation:** Seungyoo Jeon.

**Visualization:** Seungyoo Jeon.

**Writing – original draft:** Seungyoo Jeon.

**Writing – review & editing:** Seungyoo Jeon.

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
