## [Decision Letter · Decision Letter 0]

5 May 2022

PONE-D-21-22695

Measuring the Impact of Social-distancing, Testing, and Undetected Asymptomatic Cases on the Diffusion of COVID-19

PLOS ONE

Dear Dr. Jeon,

Thank you for submitting your manuscript to PLOS ONE. After careful consideration, we feel that it has merit but does not fully meet PLOS ONE’s publication criteria as it currently stands. Therefore, we invite you to submit a revised version of the manuscript that addresses the points raised during the review process.

Based on comments by the two reviewers, please expand your literature review, paying particular attention to comparison against previous work, and your discussion, specifically on limitations of your study.

We look forward to receiving your revised manuscript.

Kind regards,

Siew Ann Cheong, Ph.D.

Academic Editor

PLOS ONE

Journal Requirements:

4. Please upload a new copy of Figure 5 as the detail is not clear. Please follow the link for more information: https://blogs.plos.org/plos/2019/06/looking-good-tips-for-creating-your-plos-figures-graphics/" https://blogs.plos.org/plos/2019/06/looking-good-tips-for-creating-your-plos-figures-graphics/

5. Please ensure that you refer to Figures 1-5 in your text as, if accepted, production will need this reference to link the reader to the figure.

6. We note you have included a table to which you do not refer in the text of your manuscript. Please ensure that you refer to Tables 1-3 in your text; if accepted, production will need this reference to link the reader to the Table.

7. Please upload a copy of Supporting Information Figures S1-S5 and S1-S3 Table which you refer to in your text on page 23.

Reviewers' comments:

Reviewer's Responses to Questions

**Comments to the Author**

1. Is the manuscript technically sound, and do the data support the conclusions?

Reviewer #1: Yes

Reviewer #2: Yes

2. Has the statistical analysis been performed appropriately and rigorously? 

Reviewer #1: I Don't Know

Reviewer #2: Yes

3. Have the authors made all data underlying the findings in their manuscript fully available?

Reviewer #1: Yes

Reviewer #2: Yes

4. Is the manuscript presented in an intelligible fashion and written in standard English?

Reviewer #1: Yes

Reviewer #2: No

5. Review Comments to the Author

Reviewer #1: Dear authors

I have two main concerns about this manuscript:

1. The literature is neo clear as well as the critical analysis of the previous studies>

2. no comparison with the literature has been presented.

Kindly address these comments so I can review the manuscript again.

regards

Reviewer #2: Overall, this is a clear, concise, and well-written manuscript. The introduction is relevant and theory based. The methods are generally appropriate

As the focus was on the positive points in the work, the weaknesses should be pointed out as well and The conclusion needs an explanation of work limitations

6. PLOS authors have the option to publish the peer review history of their article (what does this mean?). If published, this will include your full peer review and any attached files.

Reviewer #1: No

Reviewer #2: **Yes: **rula a.hamid

---

## [Author Response · Author response to Decision Letter 0]

18 Jul 2022

Dear reviewers,

Thanks for your valuable feedback.

Please refer to the rebuttal letter.

---

## [Decision Letter · Decision Letter 1]

10 Aug 2022

Measuring the impact of social-distancing, testing, and undetected asymptomatic cases on the diffusion of COVID-19

PONE-D-21-22695R1

Dear Dr. Jeon,

We’re pleased to inform you that your manuscript has been judged scientifically suitable for publication and will be formally accepted for publication once it meets all outstanding technical requirements.

Kind regards,

Siew Ann Cheong, Ph.D.

Academic Editor

PLOS ONE

Additional Editor Comments (optional):

Reviewers' comments:

Reviewer's Responses to Questions

**Comments to the Author**

1. If the authors have adequately addressed your comments raised in a previous round of review and you feel that this manuscript is now acceptable for publication, you may indicate that here to bypass the “Comments to the Author” section, enter your conflict of interest statement in the “Confidential to Editor” section, and submit your "Accept" recommendation.

Reviewer #1: All comments have been addressed

Reviewer #2: All comments have been addressed

2. Is the manuscript technically sound, and do the data support the conclusions?

Reviewer #1: Yes

Reviewer #2: Yes

3. Has the statistical analysis been performed appropriately and rigorously? 

Reviewer #1: Yes

Reviewer #2: Yes

4. Have the authors made all data underlying the findings in their manuscript fully available?

Reviewer #1: Yes

Reviewer #2: Yes

5. Is the manuscript presented in an intelligible fashion and written in standard English?

Reviewer #1: Yes

Reviewer #2: Yes

6. Review Comments to the Author

Reviewer #1: Dear authors

All comments and concerns have been addressed in the revised version.

Congratulations

Regards

Reviewer #2: Overall, this is a clear, concise, and well-written manuscript. The introduction is relevant and theory based. The methods are generally appropriate

7. PLOS authors have the option to publish the peer review history of their article (what does this mean?). If published, this will include your full peer review and any attached files.

Reviewer #1: No

Reviewer #2: **Yes: **Rula a.Hamid

---

## [Editor Report · Acceptance letter]

15 Aug 2022

PONE-D-21-22695R1 

Measuring the impact of social-distancing, testing, and undetected asymptomatic cases on the diffusion of COVID-19 

Dear Dr. Jeon:

I'm pleased to inform you that your manuscript has been deemed suitable for publication in PLOS ONE. Congratulations! Your manuscript is now with our production department. 

Kind regards, 

on behalf of

Dr. Siew Ann Cheong 

Academic Editor

PLOS ONE